# Classical and Alternative Activation of Rat Microglia Treated with Ultrapure *Porphyromonas gingivalis* Lipopolysaccharide In Vitro

**DOI:** 10.3390/toxins12050333

**Published:** 2020-05-19

**Authors:** Zylfi Memedovski, Evan Czerwonka, Jin Han, Joshua Mayer, Margaret Luce, Lucas C. Klemm, Mary L. Hall, Alejandro M. S. Mayer

**Affiliations:** 1Chicago College of Osteopathic Medicine, Midwestern University, Downers Grove, IL 60515, USA; zmemedovski89@midwestern.edu (Z.M.); eczerwonka79@midwestern.edu (E.C.); 2Biomedical Sciences Program, College of Graduate Studies, Midwestern University, Downer Grove, IL 60515, USA; lklemm@wisc.edu; 3College of Dental Medicine Illinois, Midwestern University, Downers Grove, IL 60515, USA; jhan55@midwestern.edu (J.H.); jvdm@upenn.edu (J.M.); margaret.luce88@gmail.com (M.L.); 4Department of Pharmacology, College of Graduate Studies, Midwestern University, Downers Grove, IL 60515, USA; mhall1@midwestern.edu

**Keywords:** microglia, *Porphyromonas gingivalis*, *Escherichia coli*, lipopolysaccharide, neuroinflammation, periodontitis

## Abstract

The possible relationship between periodontal disease resulting from the infection of gingival tissue by the Gram-negative bacterium *Porphyromonas gingivalis* (*P. gingivalis*) and the development of neuroinflammation remains under investigation. Recently, *P. gingivalis* lipopolysaccharide (LPS) was reported in the human brain, thus suggesting it might activate brain microglia, a cell type participating in neuroinflammation. We tested the hypothesis of whether in vitro exposure to ultrapure *P. gingivalis* LPS may result in classical and alternative activation phenotypes of rat microglia, with the concomitant release of cytokines and chemokines, as well as superoxide anion (O_2_^−^), thromboxane B_2_ (TXB_2_), and matrix metalloprotease-9 (MMP-9). After an 18-h exposure of microglia to *P. gingivalis* LPS, the concentration-dependent responses were the following: 0.1–100 ng/mL *P. gingivalis* LPS increased O_2_^−^ generation, with reduced inflammatory mediator generation; 1000–10,000 ng/mL *P. gingivalis* LPS generated MMP-9, macrophage inflammatory protein 1α (MIP-1α/CCL3), macrophage inflammatory protein-2 (MIP-2/CXCL2) release and significant O_2_^−^ generation; 100,000 ng/mL *P. gingivalis* LPS sustained O_2_^−^ production, maintained MMP-9, tumor necrosis factor-α (TNF-α), and interleukin-6 (IL-6) release, and triggered elevated levels of MIP-1α/CCL3, MIP-2/CXCL2, and cytokine-induced neutrophil chemoattractant 1 (CINC-1/CXCL-1), with a very low release of lactic dehydrogenase (LDH). Although *P. gingivalis* LPS was less potent than *Escherichia coli* (*E. coli*) LPS in stimulating TXB_2_, MMP-9, IL-6 and interleukin 10 (IL-10) generation, we observed that it appeared more efficacious in enhancing the release of O_2_^−^, TNF-α, MIP-1α/CCL3, MIP-2/CXCL2 and CINC-1/CXCL-1. Our results provide support to our research hypothesis because an 18-h in vitro stimulation with ultrapure *P. gingivalis* LPS resulted in the classical and alternative activation of rat brain microglia and the concomitant release of cytokines and chemokines.

## 1. Introduction

Periodontal disease is a multifactorial and common inflammatory condition in humans that may involve chronic infection of the gingival tissue by the Gram-negative bacteria *P. gingivalis* [1]. When the infection is severe, as in periodontitis, it has been hypothesized as a risk factor for cognitive impairment and neuropathology, including dementia and Alzheimer’s disease [2,3,4]. Initial support for this hypothesis was recently provided by Ilievski et al., who observed, for the first time, that orally applied *P. gingivalis* translocated to murine brain, and was found in astrocytes, neurons and microglia, with the concomitant generation of inflammatory cytokines, and development of neurodegeneration [5]. Furthermore, *P. gingivalis* LPS has been detected in the human brains, raising the possibility that it might activate human brain microglia [6,7].

*P. gingivalis* may release significant amounts of lipopolysaccharide (LPS), a component of Gram-negative bacteria cell walls [8] that has been proposed as “a major nexus for virulence in periodontitis” because it penetrates gingival tissues [9]. Research on the chemical nature of *P. gingivalis* LPS has been ongoing [10]. These studies have shown that, in contrast to proteobacteria, such as *E. coli*, where LPS’s lipid A moiety that is attached to the disaccharide core is hexa-acylated and binds to toll-like receptor-4 (TLR4), *P. gingivalis* LPS lipid A structures are either tetra- or penta-acylated structures that appear to interact not only with TLR4 but also TLR2, because of the presence of a putatively “contaminant protein” [9]. More recent studies using standard and ultrapure *P. gingivalis* LPS, described as “different grades of purity”, appear to support the notion that ultrapure *P. gingivalis* LPS acts exclusively through TLR4 and is capable of causing the release of TNF-α, IL-6 and MCP-1 from human whole blood cells, murine cell lines, as well as a BV2 microglia cell line in vitro, though “… only weakly by ultrapure *P. gingivalis* LPS when used at high doses” [11]. In the research described in this paper, we have exclusively used ultrapure *P. gingivalis* LPS.

A number of investigators have explored the possible role of *P. gingivalis* LPS on brain microglia activation and resulting neuropathology [12,13]. Resting brain microglia have been described to proceed by either the pro-inflammatory/classical microglia activation pathway, or the anti-inflammatory/alternatively microglia activation pathway [14,15], a paradigm that has recently been reviewed [16]. Pro-inflammatory/classical activated microglia may result from infectious diseases or LPS stimulation [17,18] and are hypothesized to be involved in brain inflammation and neurodegeneration. Once activated, pro-inflammatory/classical microglia release inflammatory mediators, which may include reactive oxygen species, e.g., O_2_^−^, [17], MMP-9 [17], TXB_2_ [17] as well as cytokines TNF-α [17] and IL-6, and the chemokines CINC-1/CXCL-1, MIP-1α/CCL3, and MIP-2/CXCL-2 [19], inflammatory mediators that were studied in this research project with *P. gingivalis* LPS. In contrast, the anti-inflammatory/alternative microglia activation [15] is associated with the release of the cytokine interleukin-10 (IL-10) [19], which has been shown to be involved in tissue repair in neuroinflammatory conditions [15]. In our studies, we investigated anti-inflammatory/alternative type rat microglia activation with *P. gingivalis* LPS by assessing the generation of the cytokine IL-10 [19].

The purpose of our investigation was to experimentally test our hypothesis that the exposure of neonatal rat microglia to ultrapure *P. gingivalis* LPS in vitro would result in pro-inflammatory/classical and/or anti-inflammatory/alternatively microglia activation, and the release of pro-inflammatory and anti-inflammatory mediators. Our data provide strong experimental evidence for the proposed working hypothesis, because ultrapure *P. gingivalis* LPS activated both pro-inflammatory/classical and/or anti-inflammatory/alternatively microglia phenotypes in vitro, and, while less potent than *Escherichia coli* (*E. coli*) LPS in stimulating TXB_2_, MMP-9, IL-6 and interleukin 10 (IL-10) generation, it appeared more efficacious in enhancing release of O_2_^−^, TNF-α, MIP-1α/CCL3, MIP-2/CXCL2 and CINC-1/CXCL-1.

## 2. Results

### 2.1. Effect of P. gingivalis LPS on Neonatal Rat Brain Microglia O_2_^−^ Generation

Neuronal injury via oxidative stress as a result of reactive oxygen species generated by microglia have been thought to be involved in neurodegenerative diseases [15,17,20]. We have shown that *E. coli* LPS treatment of rat microglia will enhance O_2_^−^ generation in a concentration-dependent manner in vitro [17]. O_2_^−^ generation was determined in microglia tissue culture supernatants, as described in the Materials and Methods section. As shown in Figure 1, panel A, neonatal rat microglia released O_2_^−^ in a concentration-dependent manner when treated with either *E. coli* or *P. gingivalis* LPS for 18 h. Maximal and statistically significant O_2_^−^ release was observed at both 5 × 10^4^ to 10^5^ ng/mL *P. gingivalis* LPS. In contrast, with *E. coli* LPS, the positive control used in these experiments, rat microglia showed maximal O_2_^−^ release at 0.1 and 1 ng/mL, as shown previously [17]. Thus, in our study, *P. gingivalis* LPS was 10,000-fold less potent than *E. coli* LPS in stimulating O_2_^−^ generation from neonatal rat microglia in vitro.

### 2.2. Effect of P. gingivalis LPS on Neonatal Rat Brain Microglia LDH Generation

To determine whether an 18-h treatment with either *P. gingivalis* or *E. coli* LPS in vitro was toxic to rat neonatal microglia, we determined LDH presence in microglia culture supernatants as described in the Materials and Methods section. As shown in Figure 1, panel B, supernatants of microglia treated with *P. gingivalis* LPS (0.1–10^5^ ng/mL) did not yield significant LDH release when compared with untreated microglia (0). In contrast, although not statistically significant, *E. coli* LPS (1–100 ng/mL) treated microglia supernatants revealed a concentration-dependent increase in LDH release. Thus, *P. gingivalis* LPS did not appear to affect the microglia cell membrane integrity in the concentration range (0.1–10^5^ ng/mL) used in the in vitro experiments, but was indeed able to elicit microglia activation during the 18-h treatment, and the concomitant generation of O_2_^−^ (Figure 1), MMP-9, cytokines and chemokines (Figure 2, Figure 3 and Figure 4).

### 2.3. Effect of P. gingivalis LPS on Neonatal Rat Brain Microglia TXB_2_ Generation

TXB_2_ is an eicosanoid shown to be produced by LPS-activated microglia and hypothesized to be implicated in neuroinflammation [17,21,22]. TXB_2_ release was determined in microglia tissue culture supernatants by ELISA (see the Materials and Methods section). As shown in Figure 1, panel C, microglia treated with *P. gingivalis* LPS for 18 h showed no significant release of TXB_2_ at any of the concentrations tested in this study. In contrast, microglia treatment with *E. coli* LPS resulted in TXB_2_ production, which was statistically significant at 100 ng/mL, as previously reported [17].

### 2.4. Effect of P. gingivalis LPS on Neonatal Rat Brain Microglia MMP-9 Generation

MMP-9 is a class of zinc-metalloproteinase which is involved in extracellular matrix degradation and has been reported to be generated by rat microglia stimulated with either *E. coli* or *V. vulnificus* LPS in vitro [17,22,23,24]. MMP-9 production was determined in microglia tissue culture supernatants by ELISA (see the Materials and Methods section). As shown in Figure 2, panel A, *P. gingivalis* LPS-treated microglia released statistically significant levels of MMP-9 at 10^5^ ng/mL LPS, the concentration at which maximal O_2_^−^ release was also observed (Figure 1, panel A). In contrast, after *E. coli* LPS treatment, microglia released statistically significant MMP-9 at 1 ng/mL LPS. Thus, *P. gingivalis* LPS was 100,000-fold less potent than *E. coli* LPS in activating statistically significant MMP-9 production.

### 2.5. Effect of P. gingivalis LPS on Neonatal Rat Brain Microglia TNF-α and IL-6 Generation

The pro-inflammatory cytokine TNF-α has been shown to be involved in neurodegenerative diseases [15]. We and others have reported that LPS-stimulated microglia release TNF-α in vitro [17,22,25,26]. TNF-α release was assessed in microglia tissue culture supernatants with a Milliplex MagPix Multiplex Array (see the Materials and Methods section). As shown in Figure 2, panel B, after *P. gingivalis* LPS stimulation in vitro microglia released statistically significant TNF-α at 10^5^ ng/mL. In contrast, upon *E. coli* LPS treatment, microglia showed statistically significant TNF-α release at 10 ng/mL LPS. Thus, *P. gingivalis* LPS was observed to be 10,000-fold less potent than *E. coli* LPS in stimulating statistically significant TNF-α production.

The pro-inflammatory cytokine IL-6 has been observed in cellular survival, stress responses, and neuroinflammation [27]. We and others have reported that LPS-stimulated microglia produce IL-6 in vitro [22,25,26]. IL-6 release was assessed in microglia tissue culture supernatants with a Milliplex MagPix Multiplex Array (see the Materials and Methods section). As presented in Figure 2, panel C, maximal IL-6 release was observed upon exposure to *E. coli* LPS (10 ng/mL) and *P. gingivalis* LPS (10^5^ ng/mL), a similar concentration that triggered maximum TNF-α generation (Figure 2, panel B). However, the magnitude of IL-6 release was several-fold greater than TNF-α. Thus, *P. gingivalis* LPS in our study was 10,000-fold less potent than *E. coli* LPS in stimulating rat microglia in vitro to generate the cytokines TNF-α and IL-6.

### 2.6. Effect of P. gingivalis LPS on Neonatal Rat Brain Microglia MIP-1α/CCL3, MIP-2/CXCL-2 and CINC-1/CXCL-1 Generation

The pro-inflammatory chemokine MIP-1α/CCL3 appears to be involved in granulocyte recruitment to damaged brain regions [28]. We and others have reported that *E. coli* LPS stimulated microglia release MIP-1α/CCL3 in vitro [22,23,29,30]. MIP-1α/CCL3 release was determined in microglia tissue culture supernatants with a Milliplex MagPix Multiplex Array (see the Materials and Methods section). As shown in Figure 3, panel A, *P. gingivalis* LPS-stimulated microglia released statistically significant MIP-1α/CCL3 between 10^4^ and 10^5^ ng/mL compared to untreated controls. In contrast, *E. coli* LPS-stimulated microglia showed a statistically significant release of MIP-1α/CCL3 between 1 and 100 ng/mL. Thus, *P. gingivalis* LPS when compared with *E. coli* LPS appeared 10,000-fold less potent in stimulating statistically significant MIP-1α/CCL3 production.

The pro-inflammatory chemokine MIP-2/CXCL-2 has been reported as a chemoattractant and activator of polymorphonuclear granulocytes [31]. We and others have shown that *E. coli* LPS-stimulated microglia release MIP-2/CXCL-2 in vitro [22,32]. MIP-2/CXCL-2 release was determined in microglia tissue culture supernatants with a Milliplex MagPix Multiplex Array (see the Materials and Methods section). As shown in Figure 3, panel B, *P. gingivalis* LPS-stimulated microglia released statistically significant MIP-2/CXCL-2 between 5 × 10^4^ and 10^5^ ng/mL mL compared to untreated controls. In contrast, *E. coli* LPS-stimulated microglia showed a statistically significant release of MIP-2/CXCL-2 between 1 and 10 ng/mL. Thus, although *P. gingivalis* LPS stimulation of microglia was 10,000-fold less potent than *E. coli* LPS, it yielded statistically significant MIP-2/CXCL-2 production of a similar magnitude to that observed with MIP-1α/CCL3 release.

The pro-inflammatory chemokine CINC-1/CXCL-1 participates in neutrophil chemotaxis and activation [33]. We and others have shown that *E. coli* LPS-stimulated microglia release CINC-1/CXCL-1 in vitro [23,32,34,35]. CINC-1/CXCL-1 release was determined in microglia tissue culture supernatants with a Milliplex MagPix Multiplex Array (see the Materials and Methods section). As shown in Figure 3, panel C, *P. gingivalis* LPS-stimulated microglia released a statistically significant release of CINC-1/CXCL-1 at 10^5^ ng/mL. In contrast, *E. coli* LPS-stimulated microglia showed a non-statistically significant release of MIP-2/CXCL-2 between 1 and 100 ng/mL compared to the untreated controls. Thus, *P. gingivalis* LPS, though less potent than *E. coli* LPS, stimulated a statistically significant production of CINC-1/CXCL-1, which was of smaller magnitude than the release of both MIP-1α/CCL3 and MIP-2/CXCL-2.

### 2.7. Effect of P. gingivalis LPS on Neonatal Rat Brain Microglia IL-10 Generation 

The anti-inflammatory and immunosuppressive cytokine IL-10 [36] has been shown to be generated by *E. coli* LPS-treated mouse, rat, and human microglia [22,36,37,38]. IL-10 release was determined in microglia tissue culture supernatants with a Milliplex MagPix Multiplex Array (see the Materials and Methods section). As shown in Figure 4, *P. gingivalis* LPS-treated microglia only showed an increase in IL-10 generation at 10^5^ ng/mL compared to untreated controls. In contrast, *E. coli* LPS stimulated microglia released statistically significant IL-10 at 10 ng/mL compared to untreated controls.

### 2.8. Confocal Fluorescence Imaging of MMP-9, IL-6, MIP-1α/CCL3, MIP-2/CXCL-2 and CINC-1/CXCL-1 in Neonatal Rat Brain Microglia Treated with P. gingivalis and E. coli LPS

After neonatal microglia were stimulated for 18 h with either *P. gingivalis* or *E. coli* LPS, the release of MMP-9 and IL-6 (Figure 2), and MIP-1α/CCL3, MIP-2/CXCL-2 and CINC-1/CXCL-1 (Figure 3) into the tissue culture media was observed to be in the µg range. In order to determine whether MMP-9, IL-6, MIP-1α/CCL3, MIP-2/CXCL-2 and CINC-1/CXCL-1 were present in 1 ng/mL *E. coli* LPS-treated and 100,000 ng/mL *P. gingivalis* LPS-treated microglia after the 18 h in vitro exposure, we investigated the presence of these mediators in microglia by confocal fluorescence imaging (see the Materials and Methods section).

As shown in Figure 5, panels A and B, confocal fluorescence imaging confirmed the presence of remaining MMP-9 within 1 ng/mL *E. coli* LPS-treated microglia, but none in the cytosol of 100,000 ng/mL *P. gingivalis* LPS-treated microglia, suggesting all MMP-9 had been released into the tissue culture supernate after the microglia had been treated for 18 h in vitro with *P. gingivalis* LPS.

However, as shown below in Figure 6 (panels A and B), Figure 7 (panels A and B), Figure 8 (panels A and B) and Figure 9 (panels A and B), confocal fluorescence imaging confirmed the presence of IL-6, MIP-1α/CCL3 and MIP-2/CXCL-2 within the cytosol of 1 ng/mL *E. coli* LPS-treated microglia as well as in 100,000 ng/mL *P. gingivalis* LPS-treated microglia. CINC-1/CXCL-1 was not observed in *P. gingivalis* LPS-treated microglia cytosols. These observations suggested that, in contrast to MMP-9 generation, release of IL-6, MIP-1α/CCL3, MIP-2/CXCL-2 and CINC-1/CXCL-1 by microglia into the tissue culture supernates was ongoing after an 18-h treatment of microglia with either *E. coli* LPS or *P. gingivalis* LPS. Thus, we conclude that the generation of these mediators appeared to be a dynamic process that for some cytokines may require more than 18 h to be completed.

## 3. Discussion

Both neuroinflammation initiation and resolution, as a result of CNS infections [18] have been reported to be associated with microglia phenotypes described as either pro-inflammatory/classical or anti-inflammatory/alternatively activated [16]. One extensively studied in vivo and in vitro activator of microglia is LPS [8], which appears to activate microglia via the lipid A portion of the LPS macromolecule, resulting in the time- and concentration-dependent release of pro-inflammatory mediators, such as matrix metalloproteinases, metabolites of arachidonic acid, cytokines, chemokines and free radicals, such as O_2_^−^ [17,19,22]. Interestingly, the recent observation that *P. gingivalis* LPS has been detected in the human brains, putatively supports the proposed hypothesis that it might activate brain microglia [6,7].

The first aim of this research was to test the hypothesis that ultrapure *P. gingivalis* LPS might induce an in vitro pro-inflammatory/classical activation microglia phenotype. Our experimental observations appear to provide support for our hypothesis: First, and similar to *E. coli* LPS, which was used previously as a positive control [19], we observed that ultrapure *P. gingivalis* LPS-treated rat brain microglia produced O_2_^−^ in a concentration-dependent and statistically significant manner. Second, ultrapure *P. gingivalis* LPS-treated microglia produced the pro-inflammatory MMP-9, but no significant TXB_2_ production, thus contrasting with *E. coli* LPS-triggered microglia TXB_2_ release as we have previously reported [19,22]. Third, ultrapure *P. gingivalis* LPS-treated microglia released the following pro-inflammatory cytokines and chemokines in a concentration-dependent manner, with MIP-1α/CCL3 generation being the highest, followed by MIP-2/CXCL2, CINC-1/CXCL1, IL-6 and TNF-α. Fourth, although ultrapure *P. gingivalis* LPS appeared less potent than *E. coli* LPS in activating a rat pro-inflammatory/classical microglia phenotype and was less efficacious in stimulating the generation of TXB_2_, MMP-9, and IL-6, in contrast, the release of O_2_^−^, TNF-α and the chemokines MIP-1α/CCL3, MIP-2/CXCL2 and CINC-1/CXCL1 was increased when compared to their production by *E. coli* LPS-treated microglia. Thus, our experimental data supports our working hypothesis, namely that ultrapure *P. gingivalis* LPS (0.1–10,000 ng/mL) activates a rat pro-inflammatory/classical brain microglia phenotype in vitro, and, in contrast with *E. coli* LPS, *P. gingivalis* LPS does not seem to be toxic to the microglia cell in vitro, as shown by the minimal LDH release we observed in our experiments.

A second objective of this investigation was to determine whether treatment of microglia with ultrapure *P. gingivalis* LPS might result in an anti-inflammatory/alternatively activation phenotype, and anti-inflammatory cytokine IL-10 release, a cytokine that appears to be involved in tissue repair in the CNS [39]. We observed the following results: First, in contrast with *E. coli* LPS, ultrapure *P. gingivalis* LPS was less potent in triggering anti-inflammatory IL-10 generation after a 18 h in vitro, as depicted by the right shift observed in the dose-response curve that is presented in Figure 4. Second, while *E. coli* LPS induced statistically significant anti-inflammatory cytokine IL-10 generation, in contrast, the ultrapure *P. gingivalis* LPS treatment of microglia resulted in IL-10 generation, which though enhanced, was not statistically significant. Thus, additional studies will be required to demonstrate whether ultrapure *P. gingivalis* LPS may also trigger an anti-inflammatory/alternative activation phenotype in rat brain microglia in vitro. Furthermore, whether systemic *P. gingivalis* LPS may activate an anti-inflammatory/alternatively phenotype in rat brain microglia in vivo remains to be studied in future investigations.

It is of importance to discuss potentially new lines of research that have emerged from our studies of the in vitro effects of ultrapure *P. gingivalis* LPS on rat neonatal microglia. First, it would be significant to compare the biological activity of ultrapure *P. gingivalis*, which has been prepared by removing lipoprotein, with that of standard *P. gingivalis* LPS, and determine whether both pro-inflammatory/classical and/or anti-inflammatory/alternative rat microglia phenotypes are observed in vitro [11]. Second, testing whether LPS isolated from other *P. gingivalis* strains will also activate pro-inflammatory/classical and/or anti-inflammatory/alternative microglia phenotypes in vitro becomes a significant question worthy of further investigation [40]. Third, our in vitro study with ultrapure *P. gingivalis* LPS involved studying the pro-inflammatory/classical and/or anti-inflammatory/alternatively activation of *neonatal* rat brain microglia and the concomitant mediator response. Thus, determining whether *adult* rat microglia might also be activated by ultrapure *P. gingivalis* LPS would be of interest, as adult microglia have been shown to generate considerable PGE_2_ [41], and so the pro-inflammatory/classical and/or anti-inflammatory/alternative *adult* microglia activation phenotypes and concomitant pro-inflammatory and anti-inflammatory mediators released may differ from those we have observed using neonatal microglia in this investigation. Fourth, because *P. gingivalis* LPS has recently been observed in the human brain [7], and primary human microglia have been shown to release pro-inflammatory O_2_^−^, TXB_2_ and TNF-α [42] and anti-inflammatory IL-10 [43], the effect of ultrapure *P. gingivalis* LPS on the activation of *human* microglia in vitro should be investigated. Fifth, in vivo studies to determine whether systemic *P. gingivalis* LPS may result in rat brain microglia pro-inflammatory/classical and/or anti-inflammatory/alternative activation phenotypes should be considered as systemic LPS has been reported to cause brain inflammation [44,45]. Sixth, the possible effect of microglia’s circadian clock on the in vitro and/or in vivo activation of pro-inflammatory/classical and/or anti-inflammatory/alternative microglia activation phenotypes by *P. gingivalis* LPS should be considered in the design of both in vitro and in vivo experiments [46].

The proposed lines for further in vitro and in vivo research with *P. gingivalis* LPS will, in our view, help determine microglia pro-inflammatory/classical and/or anti-inflammatory/alternative activation phenotypes, and will hopefully contribute to novel therapeutic and diagnostic strategies for periodontal disease as well as the neuropathologies that have been hypothesized to implicate the Gram-negative bacterium *P. gingivalis*.

## 4. Conclusions

In conclusion, the current study, which demonstrates that in vitro ultrapure *P. gingivalis* LPS triggers both the classical and alternative activation of rat brain microglia, furthers our current understanding of *P. gingivalis* LPS’s potential toxicity to the brain immune system.

## 5. Materials and Methods

### 5.1. Chemicals

*Escherichia coli* LPS (*Ec*) (026:B6) was purchased from Difco Laboratories, Detroit, Mich.; ultrapure *P. gingivalis* LPS (catalog: tlrl-pglps, lot: PPG 38-01) was purchased from InvivoGen (San Diego, CA, USA). Dulbecco’s modified Eagle medium (DMEM) with high glucose (4.5 mg/L), Hanks’ balanced salt solution (HBSS), penicillin (P), streptomycin (S), trypsin (0.25%)-EDTA (1 mM) were purchased from GIBCO Laboratories, Life Technologies Inc., Grand Island, N.Y.; heat-inactivated fetal bovine serum certified (FBS) was from Hyclone, Logan, UT, USA.

### 5.2. LPS Decontamination

To inactivate LPS, all glassware and metal spatulas were baked for 4 h at 180 °C. [17]. Sterile and LPS-free 225 cm^2^ vented cell culture flasks were from BD Biosciences, San Jose, CA, USA; 24-well flat-bottom culture clusters and disposable serological pipettes were from Costar^®^, Corning Inc., Corning, NY, USA. Sterile and pyrogen-free Eppendorf Biopur pipette tips were from Brinkmann Instruments, Inc., Westbury, NY, USA.

### 5.3. Isolation of Rat Neonatal Microglia

Adherence to the National Institutes of Health guidelines on the use of experimental animals and protocol approved by Midwestern University’s Research and Animal Care Committee was followed in all experiments. Rat brain neonatal microglia was harvested and characterized as described [22]. The ethic approval board name: IACUC (Institutional Animal Care and Use Committee), Midwestern University; ethic approval code: 941, approval date: 24 January 2017.

### 5.4. Activation of Microglia with LPS (Experimental Protocol)

To determine the effect of ultrapure *P. gingivalis* LPS, 1.8–2.0 × 10^5^ neonatal microglia in DMEM + 10% FBS + 1% penicillin (P) + streptomycin (S) were plated into each well of a 24-well flat-bottom culture cluster. Thereafter, some wells remained untreated shown as 0 on the x axis of Figure 1, Figure 2, Figure 3 and Figure 4 while other wells were treated with 0.1–100,000 ng/mL *P. gingivalis*, or *E. coli* LPS (0.1–100 ng/mL) for 18 h in a humidified 5% CO_2_ incubator at 35.9 °C, as neonatal rat microglia are maximally activated by *E. coli* LPS at this time point [19,22]. Upon termination of the in vitro treatment, media (1 mL) from each tissue culture well was split into two aliquots. One aliquot (0.1 mL) was used to measure LDH levels and the remaining aliquot (0.9 mL) was frozen (−80 °C) for determination of MMP-9, TXB_2_, chemokines and cytokines as described below.

### 5.5. Lactate Dehydrogenase (LDH) Assay

Cellular toxicity following neonatal rat microglia pre-incubation with either *P. gingivalis* or *E. coli* LPS was determined spectrophotometrically as described [17,47]. LDH release was expressed as a percent of total LDH released from Triton X-100 (0.1%)-treated microglia.

### 5.6. Assay for Microglia O_2_^−^ Generation

To determine whether *P. gingivalis* LPS or *E. coli* LPS-primed neonatal rat microglia were able to generate O_2_^−^ after the 18 h incubation, once the tissue media was harvested, microglia were washed twice with warm (35.9 °C) HBSS, and then stimulated with phorbol 12-myristate 13-acetate (PMA) [1 µM] for an additional 70 min at 35.9 °C. The superoxide dismutase (SOD)-inhibitable reduction of ferricytochrome C (FCC) was used to measure microglia O_2_^−^ generation [19]. In brief, spontaneous O_2_^−^ release from unstimulated microglia was measured in the presence of FCC (50 µM) and HBSS with or without SOD (700 U). All experiments were done in triplicate and in a final volume of 1 mL. Changes in the absorbance of microglia supernatants was measured at 550 nm using a Beckman DU-650 Spectrophotometer. Differences in the amount of FCC in the presence or absence of SOD were used to determine O_2_^−^ generation by using the molecular extinction coefficient of 21 × 10^3^ M^−1^cm^−1^ and expressed in nmol/70 min.

### 5.7. Assay for Microglia TXB_2_ Generation

After an 18-h preincubation of neonatal rat microglia with either *P. gingivalis* LPS or *E. coli* LPS, the production of TXB_2_ was determined in the supernatants by ELISA (Cat. # 519031, Cayman Chemical Company, Ann Arbor, MI, USA) according to the manufacturer’s protocol. Prior to the assay, samples were diluted between 1:10 and 1:100. On the assay characteristics, the Cayman Chemical Company protocol reports that the intra-assay precision of the TXB_2_ ELISA was generated from 8 reportable results across two different concentrations of analytes in a single assay (intra-assay %Coefficient of Variation (CV) 8.2–15.3), while the inter-assay precision was generated from two different concentrations of analytes across 11 different assays (inter-assay %CV 9.9–12.9). The results were expressed as pg/mL with the minimum detectable concentration being 5 pg/mL TXB_2_.

### 5.8. Assay for Microglia MMP-9 Generation

After an 18-h preincubation of neonatal rat microglia with either *P. gingivalis* LPS or *E. coli* LPS, MMP-9 release into supernatants was assessed by ELISA (Cat. # DY8174, R&D systems, Minneapolis, MN) according to the manufacturer’s protocol. Prior to the assay samples were diluted between 1:10 and 1:100. On the assay characteristics, the R&D systems protocol reports that the intra-assay precision of the MMP-9 ELISA was generated from 3 samples of known concentration tested 20 times in a single assay (intra-assay %CV 4.5–6.9), while inter-assay precision was generated from 3 samples of known concentration in 20 separate assays by at least 3 technicians (inter-assay %CV 3.9–6.9). The results were expressed as pg/mL and the minimum detectable concentration was 78.1 pg/mL.

### 5.9. Milliplex MagPix Multiplex Array

Supernatant from untreated, *P. gingivalis* LPS and *E. coli* LPS-treated microglia was added to a 96-well Milliplex MagPix plate (Cat. # RECYTMAG-65K, Millipore, Danvers, MA, USA) to assay the inflammatory cytokines TNF-α, IL-6, CINC-1/CXCL-1, MIP-1α/CCL3, MIP-2/CXCL-2, as well as the anti-inflammatory cytokine IL-10. Samples were not diluted prior to the assay. The Milliplex plate was read by the Luminex MagPix technology according to the manufacturer’s instructions. On the assay characteristics, the Millipore protocol reports that for IL-6, IL-10, TNF-α, CINC-1/CXCL-1, MIP-2/CXCL-2, MIP-1α/CCL3, the intra-assay precision was generated from 8 reportable results across two different concentrations of analytes in a single assay (intra-assay %CV 2.3, 3.8, 2.7, 5.4, 2.9 and 4.3, respectively), while inter-assay precision was generated from two different concentrations of analytes across 11 different assays (inter-assay %CV 12.7, 9.0, 10.8, 7.7, 7.7, 9.3, respectively). The data were analyzed using xPONENT software (Luminex, Austin, TX, USA). The results were expressed as pg/mL. The minimum detectable concentrations for cytokines and chemokines were the following: IL-6, 30.7 pg/mL; IL-10, 2.7 pg/mL; TNF-α, 1.9 pg/mL; CINC-1/CXCL1, 19.7 pg/mL; MIP-2/CXCL2, 11.3 pg/mL; and MIP-1α/CCL3, 0.8 pg/mL.

### 5.10. Confocal Fluorescence Imaging

Confocal immunofluorescence microscopy was used to visualize microglia (CD11b/c), MMP-9, the cytokine IL-6, and the chemokines MIP-1α/CCL3, MIP-2/CXCL-2 and CINC-1/CXCL-1. Approximately 10^4^–7.5 × 10^4^ neonatal rat microglia were seeded on clean sterile coverslips and treated for 18 h with either (A) *E. coli* LPS (1 ng/mL), or (B) ultrapure *P. gingivalis* LPS (10^5^ ng/mL). Thereafter, the cells were fixed with freshly prepared 3.7% paraformaldehyde for 30 min, and permeabilized with ice-cold 100% MeOH for 10 min at −20 °C. After a 1-h incubation with 3% BSA/0.1% Triton X-100/PBS, the cells were incubated as follows: A 1:10,000 dilution of 4′,6-diamidino-2-phenylindole (DAPI) (Cat # P36966, Thermo Fischer Scientific, Waltman, MA, USA) was used to image microglia nuclei. To detect microglia, a primary mouse anti-rat CDllb/c antibody (1:500) (Cat # MCA275R, AbD SeroTec, Raleigh, NC, USA) and an Alexa Fluor^®^ 488 conjugated donkey anti-mouse secondary antibody (1:1000)(Cat# A21202, Thermo Fischer Scientific, Waltman, MA, USA) were used. To detect MMP-9, a primary rabbit anti-rat MMP-9 antibody (1:500) (Cat # ab38898, Abcam, Cambridge, MA) was used. To detect the cytokine IL-6, a primary mouse anti-rat IL-6 antibody (1:1000) (Cat# ab9324, Abcam, Cambridge, MA) was used. To detect the chemokines MIP-1α/CCL3, MIP-2/CXCL-2 and CINC-1/CXCL-1, the following primary antibodies were used: rabbit anti-rat MIP-1α/CCL3 antibody (1:50) (Cat# ab25128, Abcam, Cambridge, MA, USA), rabbit anti-rat MIP-2/CXCL-2 (1:50) (Cat# ab9777, Abcam, Cambridge, MA, USA), and rabbit anti-rat CINC-1/CXCL-1 antibody (Cat #ab86436, Abcam, Cambridge, MA, USA). Appropriate secondary antibodies were the following: an Alexa Fluor 594 conjugated goat anti-rabbit IgG H&L antibody (1:500) (150080, Abcam, Cambridge, MA, USA), and an Alexa Fluor 594 conjugated goat anti-mouse IgG H&L antibody (1:1000) (150116, Abcam, Cambridge, MA, USA). Fluorescently labeled microglia were visualized by using a Nikon A1R+ A1+ confocal microscope at a resolution of 512 × 512, using a 60× objective lens.

### 5.11. Statistical Analysis of the Data

The data were expressed as means ± SEM of triplicate determinations of several independent experiments, as noted in the Figure legends. Data were analyzed with Prism software package version 7 (GraphPad, San Diego, CA, USA) and tested for normality with the Shapiro–Wilk normality test. Appropriate multiway analysis of variance was then performed on all sets of data. Where significant interactions were encountered, simple effects were tested with a one-way analysis of variance followed by a Dunnett’s post-hoc test. Differences were considered statistically significant at *p* < 0.05 [19].

## Figures and Tables

**Figure 1 toxins-12-00333-f001:**
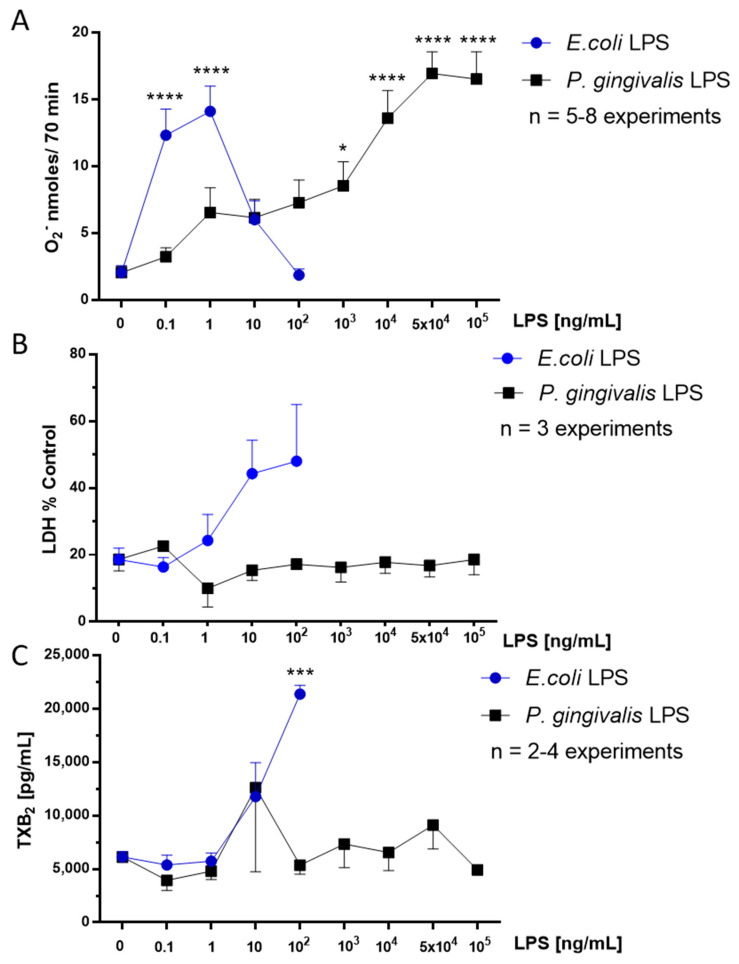
O_2_^−^ (**A**), LDH (**B**) and TXB_2_ (**C**) release by rat neonatal microglia (2.5 × 10^5^ cells/well) treated for 18 h with either *E. coli* LPS (0.1–100 ng/mL), or *P. gingivalis* LPS (0.1–10^5^ ng/mL). O_2_^−^ (**A**), LDH (**B**) and TXB_2_ (**C**) release was determined as described in the Materials and Methods section. The data are expressed as the means ± SEM of triplicate determinations from several independent experiments (n). * *p* ≤ 0.05, *** *p* ≤ 0.001, **** *p* ≤ 0.0001 LPS versus untreated control (0 ng/mL LPS).

**Figure 2 toxins-12-00333-f002:**
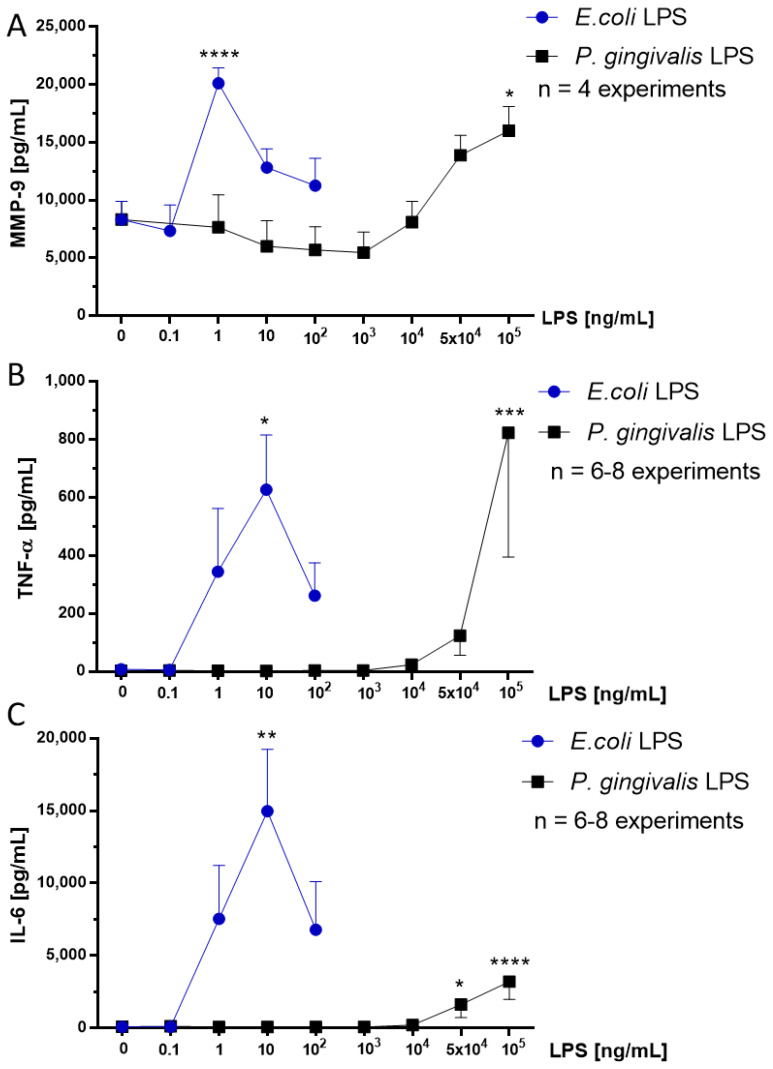
MMP-9 (**A**), TNF-α (**B**) and IL-6 (**C**) release by rat neonatal microglia (2.5 × 10^5^ cells/well) treated for 18 h with either *E. coli* LPS (0.1–100 ng/mL), or *P. gingivalis* LPS (0.1–10^5^ ng/mL). MMP-9 (**A**), TNF-α (**B**) and IL-6 (**C**) release was determined as described in the Materials and Methods section. Data expressed as pg/mL are the mean ± SEM of triplicate determinations from several independent experiments (n). * *p* ≤ 0.05, ** *p* ≤ 0.01, *** *p* ≤ 0.001, **** *p* ≤ 0.0001 LPS versus untreated control (0 ng/mL LPS).

**Figure 3 toxins-12-00333-f003:**
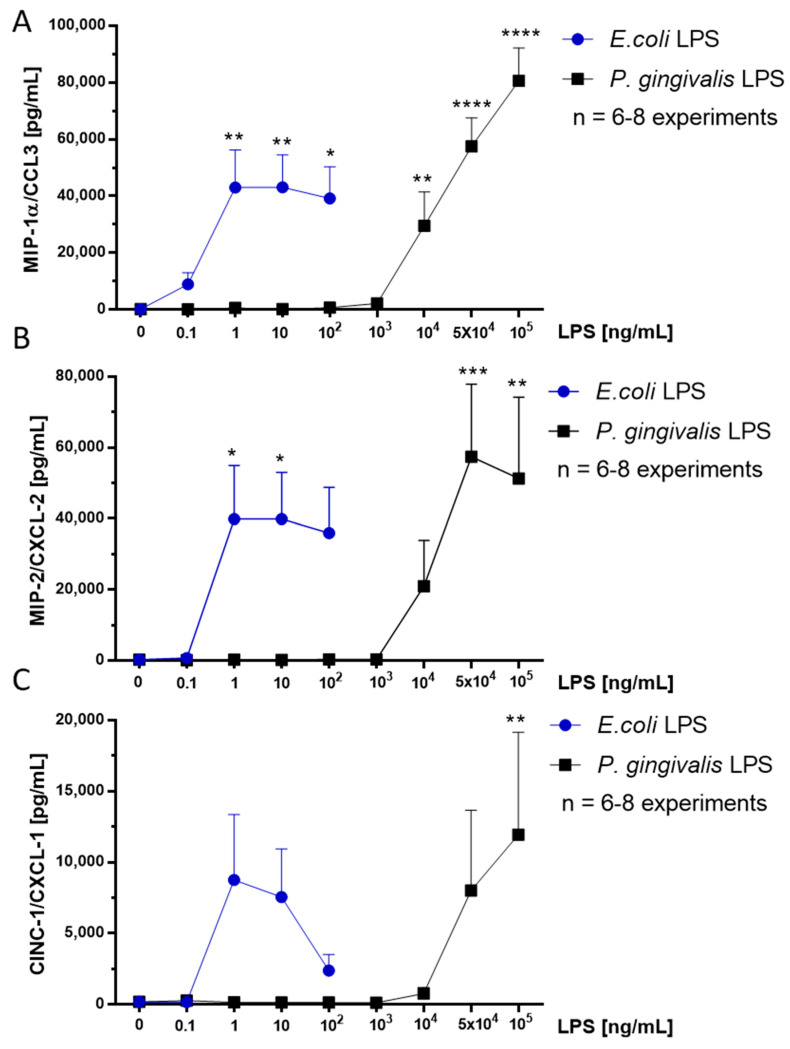
MIP-1α/CCL3 (**A**), MIP-2/CXCL-2 (**B**) and CINC-1/CXCL-1 (**C**) release by rat neonatal microglia (2.5 × 10^5^ cells/well) treated for 18 h with either *E. coli* LPS (0.1–100 ng/mL), or *P. gingivalis* LPS (0.1–10^5^ ng/mL). MIP-1α/CCL3 (**A**), MIP-2/CXCL-2 (**B**) and CINC-1/CXCL-1 (**C**) release was determined as described in the Materials and Methods section. Data expressed as pg/mL are the mean ± SEM of triplicate determinations from several independent experiments (n). * *p* ≤ 0.05, ** *p* ≤ 0.01, *** *p* ≤ 0.001, **** *p* ≤ 0.0001 LPS versus the untreated control (0 ng/mL LPS).

**Figure 4 toxins-12-00333-f004:**
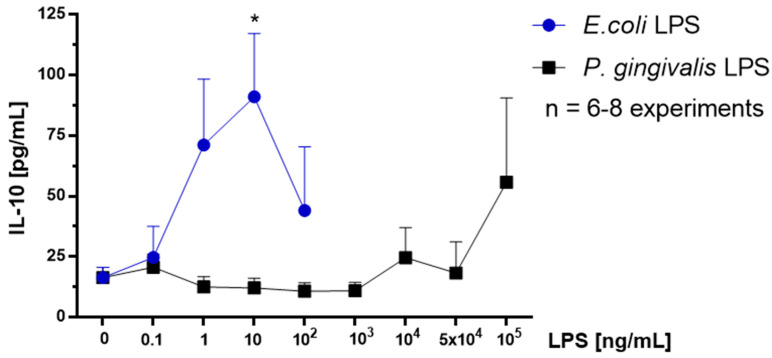
IL-10 release by rat neonatal microglia (2.5 × 10^5^ cells/well) treated for 18 h with either *E. coli* LPS (0.1–100 ng/mL), or *P. gingivalis* LPS (0.1–10^5^ ng/mL). IL-10 release was determined as described in the Materials and Methods section. Data expressed as pg/mL are the mean ± SEM of triplicate determinations from several independent experiments (n). * *p* ≤ 0.05 LPS versus the untreated control (0 ng/mL LPS).

**Figure 5 toxins-12-00333-f005:**
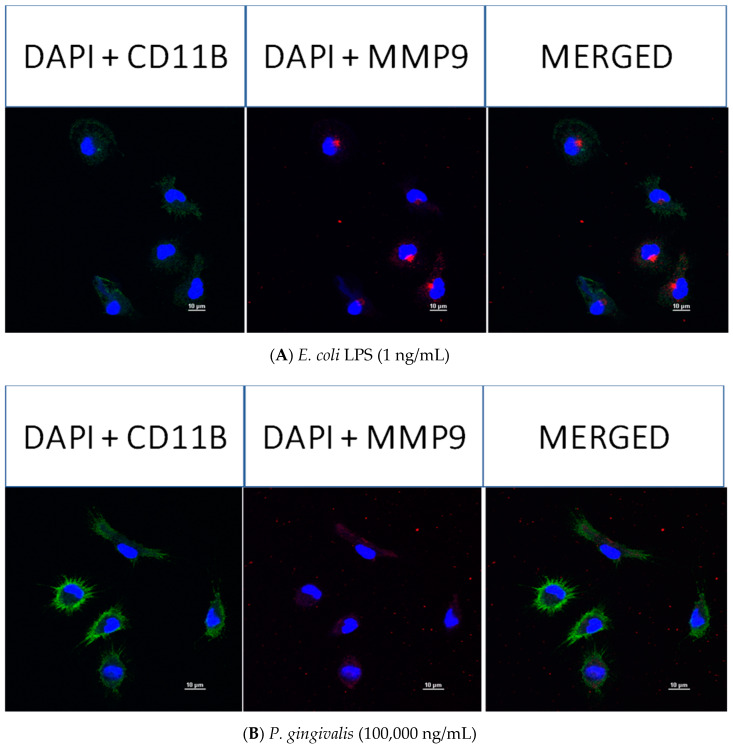
Confocal fluorescence imaging of MMP-9 in rat neonatal microglia treated for 18 h with either (**A**) *E. coli* LPS (1 ng/mL), or (**B**) *P. gingivalis* LPS (10^5^ ng/mL). The subcellular and cellular markers are 4′,6-diamidino-2-phenylindole (DAPI, blue), primary mouse anti-rat CD11b/c (green) and primary rabbit anti-rat MMP-9 (red). The scale bar denotes 10 µm. See additional details in the Materials and Methods section.

**Figure 6 toxins-12-00333-f006:**
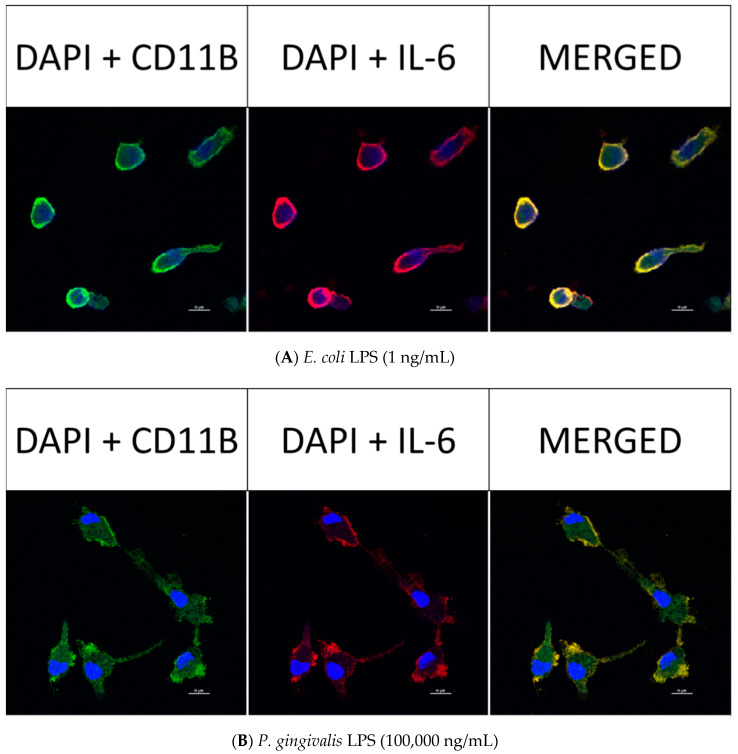
Confocal fluorescence imaging of IL-6 in rat neonatal microglia treated for 18 h with either (**A**) *E. coli* LPS (1 ng/mL), or (**B**) *P. gingivalis* LPS (10^5^ ng/mL). The cellular markers are 4′,6-diamidino-2-phenylindole (DAPI, blue), primary mouse anti-rat CD11b/c (green) and primary mouse anti-rat IL-6 (red). Scale bar denotes 10 µm. See additional details in the Materials and Methods section.

**Figure 7 toxins-12-00333-f007:**
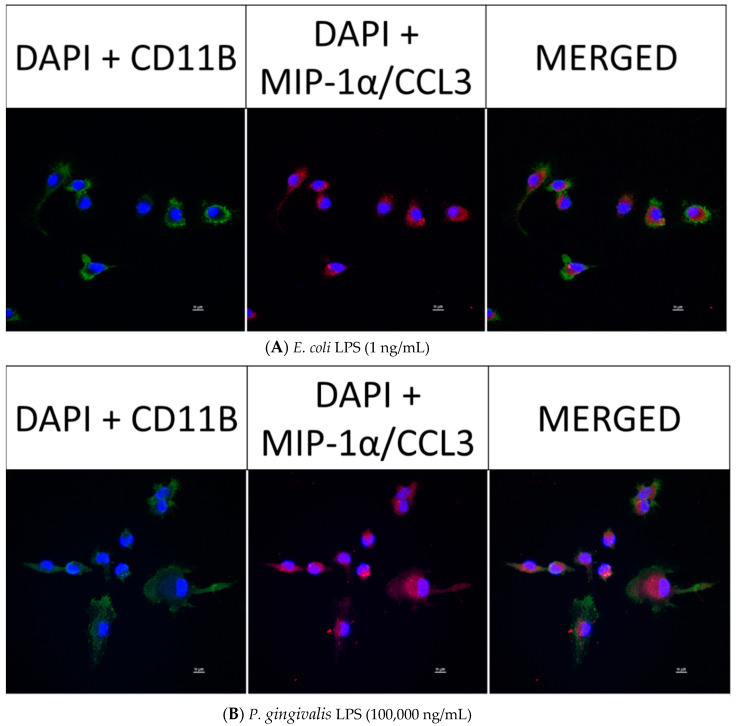
Confocal fluorescence imaging of MIP-1α/CCL3 in rat neonatal microglia treated for 18 h with either (**A**) *E. coli* LPS (1 ng/mL), or (**B**) *P. gingivalis* LPS (10^5^ ng/mL). The cellular markers are 4′,6-diamidino-2-phenylindole (DAPI, blue), primary mouse anti-rat CD11b/c (green) and primary rabbit anti-rat MIP-1α/CCL3 (red). The scale bar denotes 10 µm. See additional details in the Materials and Methods section.

**Figure 8 toxins-12-00333-f008:**
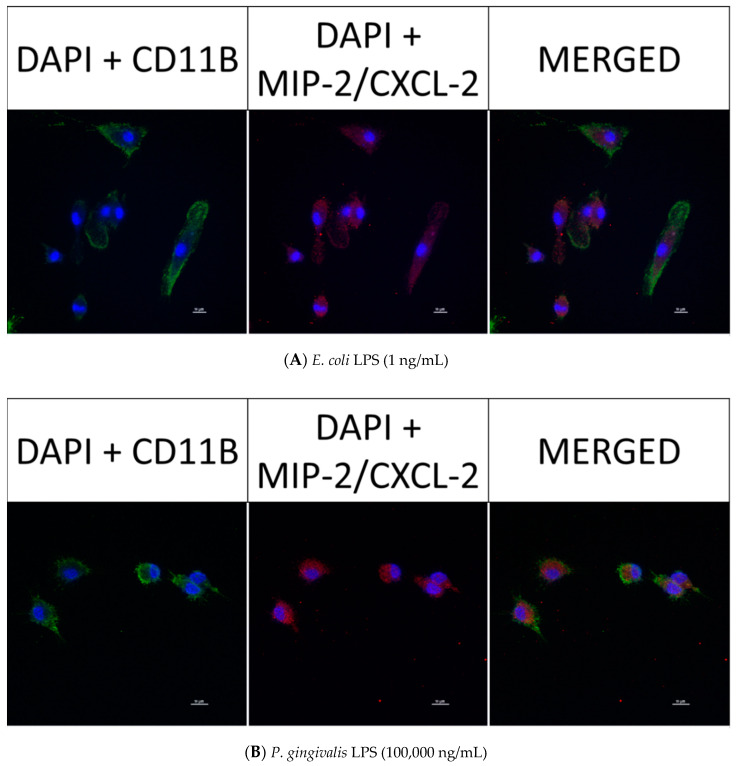
Fluorescence imaging of MIP-2/CXCL-2 in rat neonatal microglia treated for 18 h with either (**A**) *E. coli* LPS (1 ng/mL), or (**B**) *P. gingivalis* LPS (10^5^ ng/mL). The cellular markers are 4′,6-diamidino-2-phenylindole (DAPI, blue), primary mouse anti-rat CD11b/c (green) and primary rabbit anti-rat MIP-2/CXCL-2 (red). The scale bar denotes 10 µm. See additional details in the Materials and Methods section.

**Figure 9 toxins-12-00333-f009:**
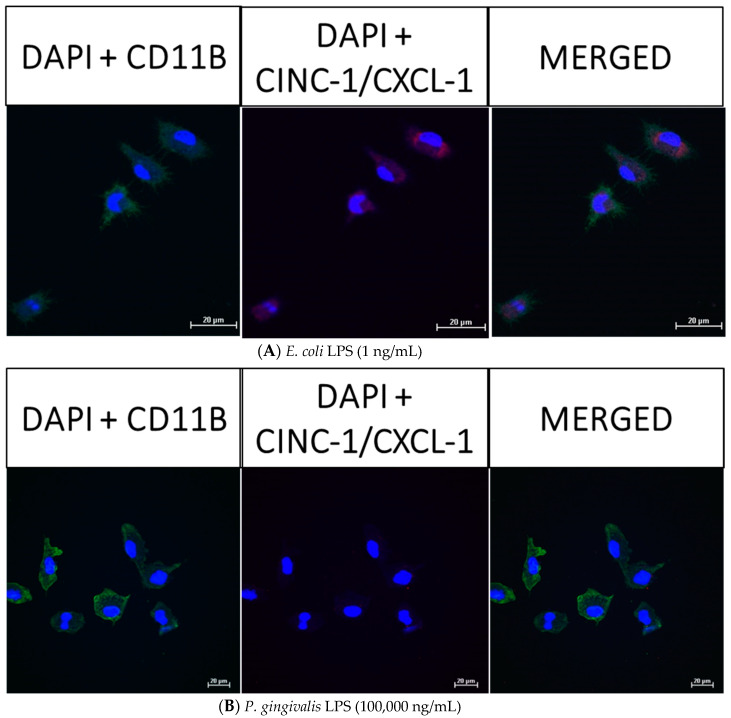
Confocal fluorescence imaging of CINC-1/CXCL-1 in rat neonatal microglia treated for 18 h with either (**A**) *E. coli* LPS (1 ng/mL), or (**B**) *P. gingivalis* LPS (10^5^ ng/mL). The cellular markers are 4′,6-diamidino-2-phenylindole (DAPI, blue), primary mouse anti-rat CD11b/c (green) and primary rabbit anti-rat CINC-1/CXCL-1 (red). The scale bar denotes 20 µm. See additional details in the Materials and Methods section.

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
