# Peer review of "Classical and Alternative Activation of Rat Microglia Treated with Ultrapure Porphyromonas gingivalis Lipopolysaccharide In Vitro"

_toxins, 2020, doi:10.3390/toxins12050333_

Round 1
Reviewer 1 Report
- It will be good to include mechanism study.
- Line 101 What is the full name of “TXB2”? Same questions for other abbreviations.
- Line 210 In Figure 5 It will be good to include the no LPS control.
- Line 395 “donkey anti-mouse CD11b secondary antibody” should be no “CD11b”.
Reviewer 2 Report
The manuscript focused on the effect of the in vitro exposure of neonatal rat microglia to ultrapure P. gingivalis LPS regarding M1-type and/or M2-type activation. The manuscript is easy to comprehend and is well written. However, I have some comments:
-In all figures it s reported n=experiments. Do the authors mean cases? It would be better to report n=cases
-The authors should explain the use of different concentration between E.Coli LPS and P.Gingivalis LPS.
Reviewer 3 Report
General Comments:
Periodontal disease has been linked to neurodegeneration through induction of neuroinflammation by LPS from the gram-negative bacterium P. gingivalis. The direct effect of LPS on rat microglia, the main CNS immune effector cell, has not previously been explored. Thus, this study aims to evaluate the effect of LPS derived from P. gingivalis on neonatal rat microglial activation. This study is timely and relevant given the number of individuals affected by age-related neurodegeneration and cognitive impairment. While I enjoyed the topic of this study, there are several items that need to be addressed before publication.
Major revisions:
- What is a relevant CNS concentration of LPS from P. gingivalis? This is not referenced in the manuscript. Your data demonstrates that very high concentrations are necessary to exhibit an effect on microglial activation. Is this data biologically relevant?
- Multiple antibodies used for immunofluorescence studies are not validated per the manufacturers website (MMP-9, CXCL1). Positive and negative controsl should be shown as a supplemental figure. Consider western blotting to demonstrate antibody specificity.
Minor revisions:
Introduction:
Line 30: Break this sentence up into 2 for reading ease
Line 31: ‘Gram-negative’ should not be capitalized. Please fix throughout manuscript.
Line 37: While references support P. gingivalis LPS in human AD brains, they do not evaluate microglial activation. Reference 6 explicitly states that RgpB-IR does not co-localize with microglia and general microglia activation was not assessed in this study. Please take this statement out with these particular references.
Line 48: These molecules were released from all cell types as you mention, but mouse cell lines, including BV-2 had a much weaker cytokine release. The way that it is written is misleading.
Line 50: Out of place – this should be addressed in methods.
Line 51: The M1/M2 terminology has been recently debated. Current argument suggests that this nomenclature oversimplifies microglial phenotype and is likely incorrect. Moreover, a single microglial cell can simultaneously express ‘M1’ and ‘M2’ markers (Morganti, 2016). Therefore, we should refrain from perpetuating M1/M2 terminology in the literature. While I understand you are summarizing others’ work, perhaps just refer to pro-inflammatory/classical or anti-inflammatory/alternatively activated microglia.
https://www.nature.com/articles/nn.4338
https://www.ncbi.nlm.nih.gov/pmc/articles/PMC4726527/
Line 59 and 62: Again, these sentences read out of place to mention what was done in the current study – this should be addressed in methods.
Line 65: As stated, your hypothesis is not directly testable. By stating that you hypothesize LPS will elicit classical and/or alternative activation, you are not defining a hypothesis. Line 67 would suggest that you hypothesize that LPS will generally activate microglia, without specific polarization toward an activation state (see previous comment regarding M1/M2 nomenclature). The release of the ‘inflammatory mediators’ is part of the activation state and does not need to specifically stated here.
Methods:
Line 345: Your reference used 17 hours, but you cite 18 hours. Is there a discrepancy? Why? I’m assuming you incubated at 37C with 5% CO2 – please clarify here.
Line 348: Define LDH
Line 353: What is the Triton X-100 group? Is this a control population not stated in section 5.4?
Line 357: Define PMA
Section 5.6: This section would read better if you used similar language to previously publication: ‘O2− generation was determined by the superoxide dismutase (SOD)-inhibitable reduction of ferricytochrome C (FCC)’.
Sections 5.7 & 5.8: How did you dilute your samples? Were samples run in duplicate or triplicate? Were all samples run at the same time? What is the inter and intraobserver variability of the plate?
Section 5.9: How did you dilute your samples? Were samples run in duplicate or triplicate? Were all samples run at the same time? What is the inter and intraobserver variability of the plate?
Line 386: extra space after visualize
Section 5.10: please state antibody concentrations
Stats: Were data tested for normal distribution?
Results
For all graphs with concentrations on y-axis: Your data is discussed in the text as ng/mL, but the y-axes in most graphs are pg/mL. To make the y-axes cleaner and consistent with the text, data should be presented as ng/mL.
Line 96-98: This is an interpretation of results and should be in the discussion, not presented here.
Line 136-139: This reads confusing. Reword please.For example: The maximal release of IL-6 was observed upon exposure to 10ng/mL (E. coli LPS) and 105 ng/mL(P. gingivalis LPS), similar to TNF-alpha release. However, the magnitude of IL-6 release was several-fold greater than TNF-alpha.
Line 148-149: The wording is incorrect here. Significance is relative to untreated controls and should be stated as such. Ex, ‘microglia had a significant increase in CCL3 release compared to untreated following exposure to LPS from P. gingivalis’.
Line 154: LPS from which bacteria? Please clarify.
Line 157-159: See previous comment for Line 148-149.
Line 163: LPS from which bacteria? Please clarify.
Line 166-167: See previous comment for Line 148-149.
Line 168-170: You don’t present your data clearly or correctly here. Your data show that CXCL-1 was detectable in the supernatant following exposure to LPS from E. coli at 1, 10, 100 ng/mL but it was not increased from untreated cells per your statistical analysis. Conversely, CXCL-1 was increased in the supernatant of microglia exposed to 105 ng/mL LPS from P. gingivalis relative to untreated cells.
Line 183-184: There is not concentration dependent increase in IL-10 release from microglia exposed to LPS from P. gingivalis. There is no difference in release at any concentration. While the mean following exposure to 105 ng/mLLPS is higher, there is tremendous variability (error bars are SEM). Thus, you cannot make this claim.
Line 185: See previous comment for Line 148-149.
Line 196-204: What do you mean by ‘release of these mediators has been completed’? Microscopy, as performed in this study, cannot give you kinetic information. It can only tell you if a protein is present at detectable levels. The kinetics of production/release/turnover cannot be elucidated from this study. Therefore use of the term ‘remaining’ is misleading and incorrect. You cannot discuss kinetics here. In the results, state where you find positive immunolabeling. You can surmise what might be happening over time in the discussion section.
References:
Line 431: small-molecule is spelled incorrectly
Round 2
Reviewer 1 Report
- Line 22-23 “... it was more efficacious in enhancing release of O2-, TNF-α, MIP-1α/CCL3, MIP-2/CXCL2 and CINC- 1/CXCL-1”, how to get this conclusion?
- Line 95 “18 h” and Line 103 “18-h” are not the same style.
- Line 103 What are the microglia activation markers? There was no significant change of TXB2 after 18 h P. gingivalis LPS treatment in figure 1, how to get the conclusion: “… indeed able to elicit microglia activation…” here?
- Line 201 “in Figure 2;” should be “in Figure 2,”; line 202 “P. gingivalis or …” should be “P. gingivalis LPS or …”.
- Line 208 – 212 causes confusion.
- Line 219 “Cellular markers” should be “Cellular nuclear markers”; in Figure 5 “CD11B” is not the same as in figure legend “CD11b/c”. Same conditions happened other places.
- In Figure 9 why there is no CINC-1/CXCL-1 positive staining in P. gingivalis LPS treatment group?
- Line 285-286 “MIP-1α/CCL3>MIP-2/CXCL2 > CINC-1/CXCL1> IL-6 > TNF-α”, how to get the conclusion?
- Line 351 “To inactivate LPS, all glassware and metal spatulas were baked for 4 h at 180°C” causes confusion.
- Line 359 “+ P + S” causes confusion.
- Line 360 “0 in x axis of” causes confusion.
Reviewer 3 Report
Thank you for your responses to the suggested changes. While most of my concerns have been addressed, there are two major points which need to be addressed before publication.
1) Section 5.7-5.9: If the samples were run on different plates at different times, it is critical to know the inter-observer variability of the plates. If this is not stated by the manufacturer, it must be tested in the laboratory. Without this information, there is no validation of your results. Moreover, dilutions should be included in the methods for the experiment to be recapitulated by others.
2) You must test your data for normal distribution. Your reply to this comment suggests that you have used the p value from the ANOVA to determine if you have a Gaussian distribution. Within Prism, you can run column statistics on your data and select from 3 different tests for Gaussian distribution (e.g., Shapiro-Wilk normality test). Did you perform this test? If not, please do and report this in your methods.
Round 3
Reviewer 1 Report
Line 73-74 "the concentration-dependent release of LDH, O2-, TXB2, MMP-9, the cytokines TNF-α, IL-6, IL-10,...". P.gingivalis LPS did not significantly increase IL-10 at any dose in Figure 4. Probably it is difficult to get the conclusion: "concentration-dependent release of ... IL-10".
Reviewer 3 Report
Thank you for your revisions to the manuscript. I appreciate the detail you have provided regarding the manufacturer's experiments to determine intra- and interassay variability. However, you still have not stated with the coefficient of variability is for each test. As a general guideline, to gauge the overall reliability of your immunoassay results, inter-assay %CV should be less than 15% while intra-assay %CV should be less than 10%. Please report what the manufacturer's %CV is. Once this is included, all my concerns have been addressed.
